# Amount of Protein Required to Improve Muscle Mass in Older Adults

**DOI:** 10.3390/nu12061700

**Published:** 2020-06-06

**Authors:** Doyeon Kim, Yongsoon Park

**Affiliations:** Department of Food and Nutrition, Hanyang University, 222 Wangsimni-ro, Seongdong-gu, Seoul 04763, Korea; kdy2313@hanyang.ac.kr

**Keywords:** increased protein amount, muscle mass, sexual dimorphism, older adults

## Abstract

Increased protein intake has been suggested as an effective strategy to treat age-related loss of muscle mass and function, but the amount of protein required to improve muscle and function without exercise in older adults remains unclear. Thus, this secondary data analysis aimed to assess what amount of protein from habitual protein intake was positively associated with changes in muscle mass and gait speed in older women and men. Ninety-six community-dwelling older adults consumed 0.8, 1.2, or 1.5 g/kg/day of protein and maintained their usual physical activity for 12 weeks. Increased protein intake of >0.54 g/kg/day was positively associated with changes in appendicular skeletal muscle mass (ASM)/weight (*B* = 0.591, *p* = 0.026), ASM/body mass index (*B* = 0.615, *p* = 0.023), and ASM:fat ratio (*B* = 0.509, *p* = 0.030) in older men. However, change in protein intake was not associated with change in muscle mass in older women. Additionally, change in protein intake was not associated with change in gait speed in older women and men. The present study suggested that an increased absolute protein amount of >0.54 g/kg/day from habitual protein intake was positively associated with change in muscle mass in older men.

## 1. Introduction

Sarcopenia, a common geriatric syndrome characterized by abnormal loss of muscle mass and function, is associated with a high risk for physical disability, functional dependence, poor quality of life, and death [1]. Korea is classified as one of the most rapidly aging countries worldwide [2], and prevalence of sarcopenia has been reported to be 20.8% among Korean elderly as defined by the European Working Group on Sarcopenia in Older People [3].

Sarcopenia is a major cause of frailty, a state of vulnerability to adverse outcomes characterized by unintentional weight loss, exhaustion, slowness, low physical activity, and weakness [4]. Among the frailty criteria, gait speed was most sensitive in predicting the onset of functional dependence in older adults [5]. Previous studies reported that approximately 60% of frail older people had sarcopenia, and mortality was higher in frail older people with sarcopenia than in those without sarcopenia [6,7]. As a modifiable risk factor, malnutrition, particularly inadequate dietary protein, has been shown to promote the development of sarcopenia and frailty [8,9].

A meta-analysis of epidemiologic studies reported that protein intake was inversely associated with frailty in older adults [10], and a clinical trial also showed that protein supplementation significantly improved lean mass in older adults [11]. The European Society for Parenteral and Enteral Nutrition Expert Group and PROT-AGE study group recommend protein intakes of 1.0–1.2 g/kg/day for healthy older adults and 1.2–2.0 g/kg/day for malnourished older adults with illness [12,13]. Previous clinical trials reported that a diet containing 1.5–1.6 g/kg/day of protein improved muscle mass without exercise compared with a diet containing 0.8–1.2 g/kg/day of protein in older adults, with a habitual protein intake of 0.8–1.1 g/kg/day [14,15]. However, Tieland et al. [16] showed that protein supplementation of 0.41 g/kg/day did not improve muscle mass without exercise compared with isocaloric placebo in older adults. On the other hand, a meta-analysis of clinical trials showed that an average 60% (>0.31 g/kg/day) increased protein intake from habitual protein intake improved resistance training-induced muscle mass in adults [17].

Our previous double-blind randomized controlled trial showed that protein supplementation of 1.5 g/kg/day improved muscle mass and gait speed compared with protein supplementation of 0.8 g/kg/day or 1.2 g/kg/day in undernourished pre-frail and frail older adults [14]. To the best of our knowledge, a study determining the amount of protein from habitual protein intake required to improve muscle mass and function without exercise in older adults has not been conducted yet. Therefore, this secondary data analysis aimed to assess the amount of protein from habitual protein intake required to improve muscle mass and gait speed for older women and men with the use of data from the original clinical trial [14].

## 2. Materials and Methods

### 2.1. Participants

Pre-frail or frail participants aged 70 to 85 years with risk of malnutrition were recruited at four welfare centers in Seoul, Korea, between May 2016 and August 2017. Participants were excluded if they had comorbidities such as liver or kidney failure, if they were participating in another clinical trial, or if they were unable to walk or communicate. Participants were assigned to the following three groups: 0.8, 1.2, or 1.5 g/kg/day protein group in the ratio of 1:1:1 for the 12-week intervention [14]. The 0.8 g/kg/day protein group consumed only placebo powder containing maltodextrin, and the 1.2 g/kg/day and 1.5 g/kg/day protein groups consumed a combination of whey protein and placebo powder (Korean Medical Food, Seoul, Korea) based on their usual protein intake assessed by 3-day 24-hour recall during screening. All participants were supplemented with 200 kcal/day of powder with 340 mL of corn silk tea (Kwangdong Pharmaceutical, Seoul, Korea). In addition, participants were asked to maintain their usual physical activity during a 12-week intervention. From a total of 120 participants who enrolled in the intervention study [14], 96 participants who completed the intervention study were included in the present study. Change in protein intake (g/kg/day) was analyzed as a categorical variable using tertiles in sex-specific tertiles and total older adults.

This study (KCT001923) was conducted according to the guidelines in the Declaration of Helsinki and approved by the Institutional Review Board of Hanyang University (HYI-15-228). All participants provided informed consent before enrollment in the study.

### 2.2. Data Collection

Information about age, sex, living alone, smoking status (never, former, or current), and alcohol drinking was obtained through interviews. Comorbid status was determined based on the presence of 0, 1, and ≥2 of the following diseases: hypertension, diabetes mellitus, cancer, chronic obstructive pulmonary disease, myocardial infarction, heart failure, angina, asthma, arthritis, cerebral ischemia, or renal disease [18]. Cognitive impairment was measured using the Korean Mini-Mental State Examination score of less than 24 [19]. Disabilities were defined as dependence in at least one item in the Korean activities of daily living (ADL) [20] and Korean instrumental activities of daily living (IADL) [21]. Risk of malnutrition was defined as a Mini-Nutritional Assessment score of less than 23.5 [22], and participants who met ≥1 and ≥3 of the modified Cardiovascular Health Study (CHS) frailty criteria [4] were considered pre-frail and frail, respectively. The CHS frailty criteria comprise five criteria, with scores ranging from 0 to 5 (unintentional weight loss ≥4.5 kg during the last year, exhaustion, low physical activity, gait speed, and weakness) [4]. Exhaustion was evaluated using the questions from the Center for Epidemiological Studies Depression scale. Low physical activity was calculated as the energy spent for a week based on the International Physical Activity Questionnaire. Slow walking speed was defined as <0.8 m/s from the average of the walking speed for 4 m, with 1.5 m before and after the walkway to allow for acceleration and deceleration. Low handgrip strength was measured twice at standing position using a digital hand grip dynamometer (Takei, Niigata, Japan), which was adjusted for sex and body mass index (BMI) [4].

### 2.3. Muscle Mass Measurement

Before and after the 12-week intervention, muscle mass was assessed by dual-energy X-ray absorptiometry (Hologic, Marlborough, MA, USA) after a 12-hour fast. Appendicular skeletal muscle mass (ASM) was estimated based on the sum of muscle mass estimated individually for two arms and two legs. Additionally, there were four types of skeletal muscle mass index (SMI): ASM adjusted for height (ASM/height^2^) = ASM (kg)/height (m^2^), ASM adjusted for weight (ASM/ weight, %) = ASM (kg)/weight (kg) × 100, ASM adjusted for BMI (ASM/BMI) = ASM (kg)/BMI (kg/m^2^) [23], and skeletal muscle to body fat ratio (ASM/fat ratio) = ASM adjusted for body fat mass (kg) [24]. Changes in ASM and SMI were calculated by obtaining the difference between the post-intervention value and the pre-intervention value.

### 2.4. Dietary Intake Measurement

Dietary intake was assessed by a registered dietitian using the 3-day 24-hour dietary recall before the intervention and after the intervention analyzed using the CAN-Pro version 4.0 (computer-aided nutritional analysis program, Korean Nutrition Society, Korea). Change in protein intake (g/kg/day) was calculated by obtaining the difference between the post-intervention value and the pre-intervention value. Changes in protein intake ranged from −0.39 to 1.01 g/kg/day in women and from −0.22 to 1.09 g/kg/day in men.

### 2.5. Statistical Analyses

The Statistical Package for the Social Sciences (SPSS) software version 24.0 (SPSS, Inc., Chicago, IL, USA) was used for statistical analyses, and *p*-values < 0.05 were considered as statistically significant. The Kolmogorov–Smirnov test was used to determine whether the data deviated from the normal distribution. Continuous variables were presented as mean ± standard deviation using the analysis of variance for normally distributed variables and the Kruskal–Wallis test for skewed distributed variables. The proportions of nominal variables were presented as numbers (percentages) using a chi-squared test.

Changes in ASM, SMI, and gait speed were compared by analysis of covariance (ANCOVA) for normally distributed variables or rank ANCOVA for skewed variables according the tertiles of change in protein intake (g/kg/day) after adjustment for baseline variables. The association between tertiles of change in protein intake and changes in ASM, SMI, and gait speed was calculated with multiple linear regression analysis after adjusting for the covariates. The lowest tertile of change in protein intake was set as the reference category, while the second and highest tertiles of change in protein intake were coded as dummy variables. Additionally, partial correlation was calculated to determine the association between change in protein intake and changes in ASM, SMI, and gait speed after adjusting for the covariates. In the multivariate models, the covariates showing *p*-values < 0.20 were selected as confounding factors and included in the fully adjusted model [25].

## 3. Results

### 3.1. Characteristics of Participants

The baseline characteristics of participants according to the tertiles of change in protein intake are shown in Table 1. There were no significant differences in age, BMI, ASM, SMI, gait speed, living alone, smoking, alcohol drinking, comorbidity, cognitive impairment, ADL and IADL disability, and frailty according to tertiles of change in protein intake in total older adults, women, and men, respectively. However, baseline protein intake was lower in participants with higher change in protein intake.

### 3.2. Muscle Mass and Gait Speed in Participants

Changes in SMI, such as ASM/weight, ASM/BMI, and ASM:fat ratio, were significantly higher in total older adults with higher change in protein intake after adjusting for baseline protein intake and ASM or SMI (Table 2). In addition, changes in ASM and SMI, such as ASM/height^2^, ASM/weight, and ASM/BMI, were significantly higher in men with higher change in protein intake after adjusting for baseline protein intake and ASM or SMI. However, change in protein intake was not significantly associated with changes in ASM or SMI in women.

### 3.3. Association Between Protein Intake and Muscle Mass and Gait Speed

The highest sex-specific tertile of change in protein intake was positively associated with changes in ASM/weight, ASM/BMI, and ASM:fat ratio in total older adults (mean, 0.72 g/kg/day) and men (>0.54 g/kg/day) compared with the lowest sex-specific tertile of change in protein intake after adjusting for covariates (Table 3). Cut-off of tertiles in protein intake were recalculated based on the distribution of protein intake among total older adults (Table 4). Multiple linear regression analysis showed that the highest tertile of change in protein intake (>0.56 g/kg/day) was positively associated with changes in ASM/weight, ASM/BMI, and ASM:fat ratio in total older adults and men compared with the lowest tertile of change in protein intake (≤0.11 g/kg/day) after adjusting for covariates. Additionally, partial correlation showed that change in protein intake was positively associated with changes in ASM/weight and ASM/BMI in men, but not in women (Figure 1). Change in protein intake was not significantly associated with change in gait speed in older adults, women, and men.

## 4. Discussion

The present study was a secondary data analysis showing that increased protein intake of >0.54 g/kg/day from habitual protein intake was positively associated with changes in ASM/weight, ASM/BMI, and ASM:fat ratio in older men. However, there was no association between change in protein intake and change in muscle mass in older women. Habitual protein intakes were 0.91 and 0.76 g/kg/day for older men and women, respectively, in the present study.

In the clinical study by Mitchell et al. [15], a diet containing 1.6 g/kg/day of protein improved muscle mass compared with a diet containing 0.8 g/kg/day of protein in older men, with a habitual protein intake of 1.10 g/kg/day, suggesting that additional 0.5 g/kg/day of protein was required to increase muscle mass. However, the 0.41 g/kg/day of protein supplementation [16] and cheese consumption containing 0.23 g/kg/day of protein [26] had no effect on muscle mass in frail or sarcopenic older adults. On the other hand, Ten Haaf et al. [27] showed that protein supplementation of 0.37 g/kg/day improved muscle mass compared with isocaloric placebo in older adults who were training for walking exercise. A combination of exercise and 25 g of protein ingestion increased muscle protein synthesis (MPS) rate by 200%, whereas 25 g of protein ingestion only increased MPS rate by 100% in adults [28], suggesting that the amount of protein required to improve muscle mass could be lower when combined with exercise. Kim et al. [29] showed that a combination of exercise and 6 g/d of amino acid supplementation improved muscle mass and function compared with exercise or 6 g/d of amino acid supplementation alone in sarcopenic older women. Furthermore, Moore et al. [30] reported that MPS reached a plateau after the ingestion of 0.40 and 0.24 g/kg of protein in older adults and younger adults, respectively, suggesting that older adults required more protein to improve muscle mass than younger adults. In a meta-analysis of clinical trials, an average 60% increase from habitual protein intake improved resistance training-induced muscle gain in adults [17]. According to the present study, in older men, an average 97% increase (>0.54 g/kg/day) from habitual protein intake increased muscle mass without exercise, consistently suggesting that exercise and younger age required less protein to increase muscle mass.

Previous studies reported that protein ingestion of 0.25 or 0.21 g/kg improved MPS by 56% in older men [31], but it was only 13% in older women [32]. Additionally, Smith et al. [33] showed that ingestion of a liquid meal of 220 kcal increased the muscle protein fractional synthesis rate by 80% in older men but not in older women. These studies suggested that older women required more protein to increase muscle mass than older men because of the difference in the anabolic resistance to protein feeding between men and women. Furthermore, a previous cohort study reported that >1.20 g/kg/day of protein intake was associated with lower prevalence of skeletal muscle mass decline compared with ≤1.20 g/kg/day of protein intake in older men but not in older women during a two-year follow-up period [34]. Consistent with the previous study, the present study showed that baseline muscle mass and protein intake were lower in women than men, and thus women might be needed more protein to increase muscle mass. On the other hand, Ormsbee et al. [35] showed that 84 g/day of protein supplementation, accounting for 1.03 g/kg/day of protein for women and 0.95 g/kg/day of protein for men, had a beneficial effect on exercise-induced muscle mass in both young women and men, with a habitual protein intake of 1.0–1.2 g/kg/day. These studies suggested that increased protein amount of 0.57 g/kg/day in the present study was insufficient to improve muscle mass in older women.

The other major finding of this study was that an increased protein amount of 0.54 g/kg/day from habitual protein intake had no significant effect on gait speed. Consistent with the present study, there were a few clinical trials showing that protein supplementation improved muscle mass but not gait speed [27,36,37]. Previous clinical trials reported that protein supplementation of 0.37 g/kg/day [27] and cheese consumption containing 0.25 g/kg/day of protein [37] had no effect on gait speed in older adults. Additionally, a recent meta-analysis of clinical trials reported that protein supplementation of approximately 0.26 g/kg/day did not improve resistance training-induced gait speed in community-dwelling older adults [38]. However, Kim et al. [39] showed a beneficial effect of protein supplementation of 0.56 g/kg/day on gait speed compared with control in older adults. This inconsistency between the present study and clinical trial by Kim et al. [39] could be partly due to the different baseline gait speed of participants. The present study showed that protein intake was not associated with gait speed in participants with usual gait speed of >1.0 m/s, but a gait speed <0.6 m/s was associated with protein intake based on a previous study reporting the beneficial effects of protein intake [39]. Thus, the additional protein intake could have no further effect on gait speed for participants exhibiting fast gait speed at baseline, and consequently, it was likely that a ceiling effect was observed.

The present study showed that ASM/weight and ASM/BMI but not ASM/height^2^ were significantly associated with increased protein from habitual protein intake in older men. Epidemiologic studies reported that protein intake was positively associated with ASM/weight [40] and ASM/BMI [41] but not with ASM/height^2^ [42] in older adults. Additionally, previous studies showed that ASM/BMI, but not ASM and ASM/height^2^, was associated with the higher risk for mobility impairment [43] and mortality [44] in older adults.

This study has the following strength: the participants in this study successfully completed a 12-week double-blind randomized controlled trial based on our previous study [14]. In our previous study, dietary intake and adherence to the intervention were carefully monitored by dietitians, and muscle mass was assessed by dual-energy X-ray absorptiometry, which was considered as the reference standard for measuring muscle mass. However, the present study has a few limitations. First, the sample size of older men according to tertiles of change in protein intake (g/kg/day) was small. However, the power of SMI was >70%, reliable for detecting differences. Second, the findings should be interpreted with caution because of the nature of secondary data analysis. Third, the generalizability of our findings to other populations may be limited since the participants of the present study included only undernourished frail Korean older adults. Fourth, although adjustments were made for various confounding factors, it is possible that unmeasured factors affected the results of this study.

## 5. Conclusions

The present study suggested that increased protein amount of >0.54 g/kg/day from habitual protein intake was positively associated with changes of muscle mass in older men. Further study is required to investigate the amount of protein required to improve muscle mass for older women and to confirm whether additional protein intake of >0.54 g/kg/day improves muscle mass in a clinical trial of older men.

## Figures and Tables

**Figure 1 nutrients-12-01700-f001:**
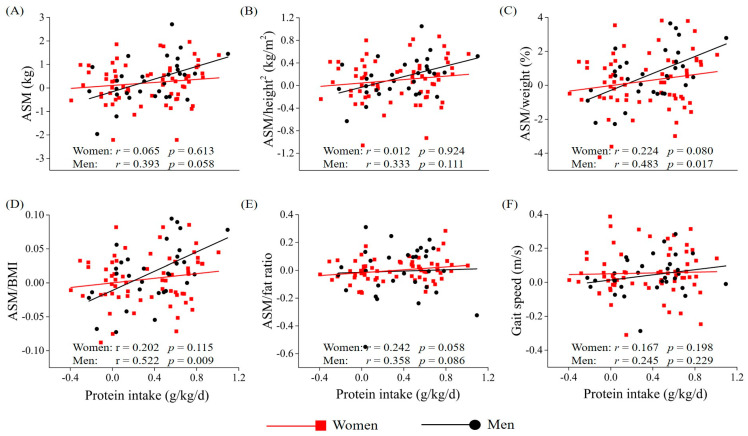
Partial correlation between changes in muscle mass (**A–E**) and gait speed (**F**) and change in protein intake (g/kg/day) in women and men. Red squares (■) represent data for women, and black circles (●) represent data for men. For partial correlation analysis to assess the relationship between muscle mass and protein intake, it was adjusted for cognitive impairment, baseline dietary intake of protein, and baseline muscle mass in women; age, alcohol drinking, cognitive impairment, frailty, activities of daily living disability, baseline dietary intake of protein, and baseline in men. For partial correlation analysis to assess the relationship between gait speed and protein intake, it was adjusted for living alone, activities of daily living disability, instrumental activities of daily living disability, and baseline gait speed in women; age, smoking, comorbidity, instrumental activities of daily living disability, and baseline gait speed in men. ASM, appendicular skeletal muscle mass; BW, body weight; BMI, body mass index.

**Table 1 nutrients-12-01700-t001:** Baseline characteristics according to sex-specific tertiles of change in protein intake.

Change in Protein Intake (g/kg/day)	Total (*n* = 96)				Women (*n* = 65)				Men (*n* = 31)			
	Tertile 1 (Mean, −0.06)	Tertile 2 (Mean, 0.36)	Tertile 3 (Mean, 0.72)	*p*	Tertile 1 (≤0.04)	Tertile 2 (0.05–0.57)	Tertile 3 (>0.57)	*p*	Tertile 1 (≤0.15)	Tertile 2 (0.16–0.54)	Tertile 3 (>0.54)	*p*
Age (years)	76.97 ± 3.78	76.79 ± 3.42	77.06 ± 3.80	0.954 ^a^	77.24 ± 3.51	76.68 ± 3.30	77.91 ± 3.80	0.520 ^a^	76.40 ± 4.45	77.00 ± 3.80	75.20 ± 3.23	0.563 ^a^
BMI (kg/m^2^)	24.37 ± 4.48	24.29 ± 2.67	23.76 ± 2.46	0.656 ^a^	25.47 ± 4.67	24.23 ± 2.73	23.97 ± 2.46	0.441 ^a^	22.05 ± 3.10	24.43 ± 2.69	23.30 ± 2.52	0.165 ^a^
ASM (kg)	14.86 ± 3.27	15.32 ± 3.59	14.78 ± 2.93	0.932 ^b^	13.41 ± 2.24	13.29 ± 1.63	13.41 ± 2.12	0.972 ^a^	17.91 ± 3.02	19.39 ± 2.91	17.80 ± 2.09	0.483 ^b^
ASM/height^2^ (kg/m^2^)	6.26 ± 0.89	6.18 ± 0.88	6.10 ± 0.76	0.856 ^b^	6.04 ± 0.85	5.79 ± 0.51	5.85 ± 0.65	0.500 ^a^	6.71 ± 0.83	6.97 ± 0.95	6.66 ± 0.71	0.657 ^a^
ASM/weight (%)	26.23 ± 4.52	25.57 ± 3.39	25.81 ± 3.28	0.873 ^b^	24.15 ± 3.53	24.08 ± 2.66	24.47 ± 2.36	0.891 ^a^	30.59 ± 3.01	28.56 ± 2.68	28.74 ± 3.20	0.247 ^a^
ASM/BMI	0.63 ± 0.17	0.63 ± 0.14	0.63 ± 0.13	0.905 ^b^	0.54 ± 0.09	0.55 ± 0.07	0.56 ± 0.07	0.555 ^a^	0.82 ± 0.12	0.79 ± 0.09	0.77 ± 0.10	0.610 ^a^
ASM:fat ratio	1.09 ± 0.51	0.98 ± 0.40	1.11 ± 0.66	0.735 ^b^	0.81 ± 0.27	0.80 ± 0.26	0.82 ± 0.16	0.979 ^a^	1.66 ± 0.39	1.35 ± 0.39	1.74 ± 0.89	0.169 ^a^
Gait speed (m/s)	0.97 ± 0.32	0.98 ± 0.32	1.00 ± 0.34	0.978 ^a^	0.91 ± 0.30	0.96 ± 0.27	0.96 ± 0.34	0.853 ^a^	1.09 ± 0.34	1.02 ± 0.41	1.05 ± 0.35	0.913 ^a^
Living alone, *n* (%)	17 (54.8)	19 (57.6)	18 (56.3)	0.976 ^c^	13 (61.9)	14 (63.6)	16 (72.7)	0.720 ^c^	4 (40.0)	5 (45.5)	2 (20.0)	0.446 ^c^
Smoking, *n* (%)				0.197 ^c^				0.380 ^c^				0.105 ^c^
Never	22 (71.0)	20 (60.6)	24 (75.0)		19 (90.5)	20 (90.9)	22 (100.0)		3 (30.0)	0 (0.0)	2 (20.0)	
Former	8 (25.8)	9 (27.3)	3 (9.4)		2 (9.5)	1 (4.5)	0 (0.0)		6 (60.0)	8 (72.7)	3 (30.0)	
Current	1 (3.2)	4 (12.1)	5 (15.6)		0 (0.0)	1 (4.5)	0 (0.0)		1 (10.0)	3 (27.3)	5 (50.0)	
Alcohol drinking, *n* (%)	23 (74.2)	21 (63.6)	22 (68.8)	0.661 ^c^	14 (66.7)	11 (50.0)	14 (63.6)	0.490 ^c^	9 (90.0)	10 (90.9)	8 (80.0)	0.717 ^c^
Comorbidity, *n* (%) ^d^				0.756 ^c^				0.992 ^c^				0.476 ^c^
0	10 (32.3)	9 (27.3)	6 (18.8)		4 (19.0)	4 (18.2)	4 (18.2)		6 (60.0)	5 (45.5)	2 (20.0)	
1	12 (38.7)	12 (36.4)	15 (46.9)		9 (42.9)	8 (36.4)	9 (40.9)		3 (30.0)	4 (36.4)	6 (60.0)	
≥2	9 (29.0)	12 (36.4)	11 (34.4)		8 (38.1)	10 (45.5)	9 (40.9)		1 (10.0)	2 (18.2)	2 (20.0)	
Cognitive impairment, *n* (%) ^e^	6 (19.4)	10 (30.3)	11 (34.4)	0.392 ^c^	5 (23.8)	9 (40.9)	6 (27.3)	0.435 ^c^	1 (10.0)	1 (9.1)	5 (50.0)	0.042 ^c^
ADL disability, *n* (%)	7 (22.6)	11 (33.3)	6 (18.8)	0.371 ^c^	4 (19.0)	7 (31.8)	5 (22.7)	0.604 ^c^	3 (30.0)	4 (36.4)	1 (10.0)	0.361 ^c^
IADL disability, *n* (%)	11 (35.5)	14 (42.4)	16 (50.0)	0.507 ^c^	5 (23.8)	8 (36.4)	8 (34.6)	0.599 ^c^	6 (60.0)	6 (54.5)	8 (80.0)	0.446 ^c^
Frailty, *n* (%)	2 (6.5)	10 (30.3)	6 (18.8)	0.051 ^c^	2 (9.5)	6 (27.3)	5 (22.7)	0.321 ^c^	0 (0.0)	4 (36.4)	1 (10.0)	0.063 ^c^
MNA score	20.32 ± 2.15	20.35 ± 2.47	21.11 ± 1.92	0.287 ^b^	20.05 ± 2.09	20.34 ± 2.33	20.98 ± 1.81	0.356 ^b^	20.90 ± 2.26	20.36 ± 2.85	21.40 ± 2.23	0.228 ^a^
Protein intake (g/kg/day)	0.93 ± 0.29 ^f^	0.79 ± 0.21 ^f^	0.72 ± 0.20 ^g^	0.008 ^a^	0.83 ± 0.29 ^f^	0.80 ± 0.20 ^g^	0.67 ± 0.13 ^g^	0.018 ^a^	1.12 ± 0.21 ^f^	0.79 ± 0.24 ^g^	0.82 ± 0.28 ^g^	0.008 ^a^

Data were presented as mean ± standard deviation or number of participants (percentage distribution), as appropriate. *p*-values for differences between tertiles of change in protein intake were analyzed using the ^a^ analysis of covariance (ANOVA) for normally distributed continuous variables, the ^b^ Kruskal–Wallis test for non-normally distributed variables, and the ^c^ chi-squared test for categorical variables. ^d^ Comorbidities including hypertension, diabetes mellitus, cancer, chronic obstructive pulmonary disease, myocardial infarction, heart failure, angina, asthma, arthritis, cerebral ischemia, and renal disease. ^e^ Korean Mini-Mental State Examination score of less than 24. ^f,g^ Different superscript letters among total older adults, women, and men within a row were significantly different by ANCOVA test with the Dunn–Bonferroni post hoc test. ASM, appendicular skeletal muscle mass; BMI, body mass index; MNA, Mini-Nutritional Assessment; ADL, activities of daily living; IADL, instrumental activities of daily living.

**Table 2 nutrients-12-01700-t002:** Changes in muscle mass and gait speed according to sex-specific tertiles of change in protein intake.

Change in Protein Intake (g/kg/day)	Total (*n* = 96)				Women (*n* = 65)				Men (*n* = 31)			
	Tertile 1 (Mean, −0.06)	Tertile 2 (Mean, 0.36)	Tertile 3 (Mean, 0.72)	*p*	Tertile 1 (≤0.04)	Tertile 2 (0.05–0.57)	Tertile 3 (>0.57)	*p*	Tertile 1 (≤0.15)	Tertile 2 (0.16–0.54)	Tertile 3 (>0.54)	*p*
ASM (kg)	0.07 ± 0.88	0.18 ± 0.73	0.51 ± 0.95	0.075 ^a^	0.14 ± 0.86	0.22 ± 0.81	0.29 ± 0.91	0.976 ^a^	−0.08 ± 0.96 ^c^	0.10 ± 0.56 ^c^	1.01 ± 0.85 ^d^	0.012 ^a^
ASM/height^2^ (kg/m^2^)	0.04 ± 0.37	0.09 ± 0.31	0.21 ± 0.39	0.174 ^a^	0.06 ± 0.39	0.09 ± 0.35	0.14 ± 0.40	0.904 ^a^	−0.01 ± 0.34 ^c^	0.07 ± 0.21 ^c^	0.38 ± 0.33 ^d^	0.019 ^a^
ASM/weight (%)	−0.06 ± 1.67 ^c^	0.32 ± 1.27 ^c^	0.86 ± 1.72 ^d^	0.026 ^a^	−0.03 ± 1.76	0.40 ± 1.47	0.39 ± 1.72	0.393 ^a^	−0.13 ± 1.57 ^c^	0.18 ± 0.78 ^d^	1.88 ± 1.26 ^d^	0.002 ^a^
ASM/BMI	−0.00 ± 0.04 ^c^	0.01 ± 0.03 ^c^	0.02 ± 0.04 ^d^	0.028 ^a^	−0.00 ± 0.04	0.01 ± 0.03	0.01 ± 0.04	0.427 ^a^	−0.01 ± 0.04 ^c^	0.00 ± 0.03 ^d^	0.05 ± 0.04 ^d^	0.003 ^a^
ASM:fat ratio	−0.04 ± 0.14 ^c^	0.02 ± 0.10 ^c^	0.01 ± 0.12 ^d^	0.033 ^a^	−0.03 ± 0.08	0.01 ± 0.07	0.01 ± 0.10	0.200 ^a^	−0.06 ± 0.23	0.02 ± 0.14	0.02 ± 0.17	0.147 ^a^
Gait speed (m/s)	0.04 ± 0.12	0.04 ± 0.14	0.07 ± 0.13	0.066 ^b^	0.06 ± 0.13	0.04 ± 0.14	0.07 ± 0.14	0.398 ^b^	0.00 ± 0.07	0.04 ± 0.14	0.07 ± 0.11	0.070 ^a^

Data were presented as mean ± standard deviation or number of participants (percentage distribution), as appropriate. *p*-values for differences between tertiles of change in protein intake (g/kg/day) were analyzed using the ^a^ analysis of covariance (ANCOVA) for normally distributed continuous variables and the ^b^ ranked ANCOVA for non-normally distributed variables after adjusting for baseline protein intake (g/kg/day) and muscle mass or gait speed. ^c,d^ Different superscript letters among total older adults, women, and men within a row were significantly different by ANCOVA test with Dunn–Bonferroni post hoc test. ASM, appendicular skeletal muscle mass; BMI, body mass index.

**Table 3 nutrients-12-01700-t003:** Multiple linear regression analysis of changes in muscle mass and gait speed according to sex-specific tertiles of change in protein intake.

Change in Protein Intake (g/kg/day)	*n*	ASM (kg)		ASM/Height^2^ (kg/m^2^)		ASM/Weight (%)		ASM/BMI		ASM:Fat Ratio		Gait Speed (m/s)	
		β	*p* ^a^	β	*p* ^a^	β	*p* ^a^	β	*p* ^a^	β	*p* ^a^	β	*p* ^b^
Total													
Tertile 1 (mean, −0.06)	31	1.00 (ref.)		1.00 (ref.)		1.00 (ref.)		1.00 (ref.)		1.00 (ref.)		1.00 (ref.)	
Tertile 2 (mean, 0.36)	33	0.074	0.509	0.016	0.888	0.119	0.301	0.148	0.201	0.207	0.066	0.006	0.947
Tertile 3 (mean, 0.72)	32	0.175	0.135	0.120	0.311	0.304	0.012	0.314	0.010	0.318	0.008	0.170	0.079
Women													
Tertile 1 (≤0.04)	21	1.00 (ref.)		1.00 (ref.)		1.00 (ref.)		1.00 (ref.)		1.00 (ref.)		1.00 (ref.)	
Tertile 2 (0.04 < to ≤ 0.57)	22	−0.007	0.960	−0.088	0.512	0.126	0.376	0.157	0.284	0.221	0.143	0.000	0.998
Tertile 3 (>0.57)	22	0.029	0.842	−0.026	0.850	0.196	0.191	0.172	0.262	0.256	0.107	0.151	0.182
Men													
Tertile 1 (≤0.15)	10	1.00 (ref.)		1.00 (ref.)		1.00 (ref.)		1.00 (ref.)		1.00 (ref.)		1.00 (ref.)	
Tertile 2 (0.15 < to ≤ 0.54)	11	0.361	0.102	0.275	0.258	0.144	0.526	0.199	0.385	0.301	0.161	0.188	0.282
Tertile 3 (>0.54)	10	0.357	0.144	0.393	0.148	0.591	0.026	0.615	0.023	0.509	0.030	0.286	0.130

^a^ For multiple linear regression analysis, it was adjusted for sex, age, cognitive impairment, activities of daily living disability, baseline dietary protein intake (g/kg/day), and baseline muscle mass in total older adults; cognitive impairment, baseline dietary protein intake (g/kg/day), and baseline muscle mass in women; age, alcohol drinking, cognitive impairment, frailty, activities of daily living disability, baseline dietary protein intake (g/kg/day), and baseline muscle mass in men. ^b^ For multiple linear regression analysis, it was adjusted for living alone, activities of daily living disability, instrumental activities of daily living disability, and baseline gait speed in total older adults and women; age, smoking, comorbidity, instrumental activities of daily living disability, and baseline gait speed in men. ASM, appendicular skeletal muscle mass; BMI, body mass index; Ref., reference.

**Table 4 nutrients-12-01700-t004:** Multiple linear regression analysis of changes in muscle mass and gait speed according to tertiles of change in protein intake based on the distribution of total older adults.

Change in Protein Intake (g/kg/day)	*n*	ASM (kg)		ASM/Height^2^ (kg/m^2^)		ASM/Weight (%)		ASM/BMI		ASM:Fat Ratio		Gait Speed (m/s)	
		β	*p* ^a^	β	*p* ^a^	β	*p* ^a^	β	*p* ^a^	β	*p* ^a^	β	*p* ^b^
Total													
Tertile 1 (≤0.11)	32	1.00 (ref.)		1.00 (ref.)		1.00 (ref.)		1.00 (ref.)		1.00 (ref.)		1.00 (ref.)	
Tertile 2 (0.11–0.56)	32	0.067	0.545	−0.002	0.984	0.104	0.359	0.143	0.210	0.113	0.315	0.033	0.732
Tertile 3 (>0.56)	32	0.171	0.142	0.109	0.355	0.293	0.014	0.307	0.010	0.262	0.026	0.183	0.059
Women													
Tertile 1 (≤0.11)	24	1.00 (ref.)		1.00 (ref.)		1.00 (ref.)		1.00 (ref.)		1.00 (ref.)		1.00 (ref.)	
Tertile 2 (0.11–0.56)	19	−0.027	0.839	−0.103	0.433	0.100	0.464	0.124	0.374	0.128	0.379	−0.003	0.975
Tertile 3 (>0.56)	22	0.019	0.894	−0.032	0.817	0.178	0.222	0.149	0.316	0.202	0.195	0.149	0.172
Men													
Tertile 1 (≤0.11)	8	1.00 (ref.)		1.00 (ref.)		1.00 (ref.)		1.00 (ref.)		1.00 (ref.)		1.00 (ref.)	
Tertile 2 (0.11–0.56)	13	0.395	0.074	0.352	0.159	0.257	0.277	0.318	0.172	0.263	0.232	0.319	0.080
Tertile 3 (>0.56)	10	0.475	0.082	0.520	0.090	0.704	0.017	0.739	0.012	0.525	0.039	0.378	0.051

^a^ For multiple linear regression analysis, it was adjusted for sex, age, cognitive impairment, activities of daily living disability, baseline dietary protein intake (g/kg/day), and baseline muscle mass in total older adults; cognitive impairment, baseline dietary protein intake (g/kg/day), and baseline muscle mass in women; age, alcohol drinking, cognitive impairment, frailty, activities of daily living disability, baseline dietary protein intake (g/kg/day), and baseline muscle mass in men. ^b^ For multiple linear regression analysis, it was adjusted for living alone, activities of daily living disability, instrumental activities of daily living disability, and baseline gait speed in total older adults and women; age, smoking, comorbidity, instrumental activities of daily living disability, and baseline gait speed in men. ASM, appendicular skeletal muscle mass; BMI, body mass index; Ref., reference.

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
