# Peer review of "Amount of Protein Required to Improve Muscle Mass in Older Adults"

_nutrients, 2020, doi:10.3390/nu12061700_

Round 1

Reviewer 1 Report

Kim D and Park Y analyzed the amount of protein intake required for the improvement of muscle mass in older adults.

Major points

1) In this study, the authors concluded that an additional protein intake of > 0.54 g/kg/day was required to increase muscle mass. However, the reasons for that conclusion were not described in the results.

2) In table 3, similar to question 1), how did the authors decide 0.11 and 0.56 g/kg/day as values to be separated? The authors should describe 3.3 in more detail.

3) In the introduction, the authors should describe the summary of their first analysis and its finding. In addition, the differences between the first and second analyses should be described.

Author Response

Major points

1) In this study, the authors concluded that an additional protein intake of > 0.54 g/kg/day was required to increase muscle mass. However, the reasons for that conclusion were not described in the results.

Response: We agree with your opinion. The highest tertile of change in protein intake among total adults was >0.56 g/kg/day, but that among men was >0.54 g/kg/day. We revised the result session stating the different tertile values in line 146-153.

2) In table 3, similar to question 1), how did the authors decide 0.11 and 0.56 g/kg/day as values to be separated? The authors should describe 3.3 in more detail.

Response: Sorry for the confusion. We revised the result session stating that change in protein intake was divided into the tertiles based on the distribution of total older adults, <0.11, 0.11-0.56, >0.56 g/kg/day (Table 4), and the tertiles based on the distribution of men, <0.15, 0.15-0.54, >0.54 g/kg/day in results, line 146-153.

3) In the introduction, the authors should describe the summary of their first analysis and its finding. In addition, the differences between the first and second analyses should be described.

Response: Thank you for your suggestion. We revised the introduction summarizing the results of the original clinical trial, and stating that the present study was the secondary analysis aimed to assess the amount of protein from habitual protein intake required to improve muscle mass and gait speed in line 49-51 and 54-56.

Reviewer 2 Report

The subject of the study is of great interest in the area, however, the statistical analysis should be reviewed.

For this secondary analysis, the authors divided the participants in six groups, according sex and terliles of changes in protein intake. Nevertheles, the baseline characteristics were presented considering solely sex (two groups) and the confounding factors to adjust the multivariate models based in the diferences found between these two groups. The diferences between the groups according the tertiles in protein intake were negleted.

Also, because sample size calculation was not performed for this secondary analysis, the authors should consider presenting the power of their results to make then trustworthy.

Taking into account the small number of men in the sample, maybe analysis separated by sex should not be considered, and only the regression with the total of the participants (adjusted for sex as performed by the authors) should be considered.

Other comments are on the text.

Author Response

For this secondary analysis, the authors divided the participants in six groups, according sex and terliles of changes in protein intake. Nevertheless, the baseline characteristics were presented considering solely sex (two groups) and the confounding factors to adjust the multivariate models based in the differences found between these two groups. The differences between the groups according the tertiles in protein intake were neglected.

Response: Thank you for your comments. We reanalyzed the baseline characteristics according to tertiles of change in protein intake among total older adults, women, and men separately.

Also, because sample size calculation was not performed for this secondary analysis, the authors should consider presenting the power of their results to make then trustworthy.

Taking into account the small number of men in the sample, maybe analysis separated by sex should not be considered, and only the regression with the total of the participants (adjusted for sex as performed by the authors) should be considered.

Response: We agree with your opinion. Sample size of older men according to tertiles of changes in protein intake was small in absolute terms. However, power of ASM/weight, ASM/BMI, and ASM:fat ratio was ≥70% which was required to achieve reliability of statistical power by Cohen (Statistical power analysis for the behavioral sciences;1998). In addition, we revised the limitation in line 262-263.

Other comments are on the text.

This is the description of the participants and criteria of allocation for the randomized controlled trial. For this secondary analysis, the participants were allocated according to sex and tertiles of changes in protein intake. Therefore the randomization was lost. Is secondary analysis of a randomized controlled trial suitable in the title? The information of how the participants were grouped for this secondary analysis (6 groups in the total) should be described in the participants section.

Response: We agree with your opinion. We deleted “Secondary analysis of a Randomized Controlled Trial” in the title. In addition, we stated the sentence “Change in protein intake (g/kg/day) was analyzed as a categorical variable using tertiles in sex-specific tertiles and total older adults.” in participant section in line 71-72.

Considering that no sample size calculation was performed for this secondary analysis, the authors should present the power or effect size of their results.

Woudn't it be more interesting to present only the analysis of the total group, without seperating men and women? I ask that because of the small number of men in the sample. Power analysis will show if it was sufficient.

Response: We agree with your opinion. Sample size of older men according to tertiles of changes in protein intake was small in absolute terms. However, power of ASM/weight, ASM/BMI, and ASM:fat ratio was ≥70% which was required to achieve reliability of statistical power by Cohen (Statistical power analysis for the behavioral sciences;1998). In addition, we revised the limitation in line 262-263.

Include the physical activity cut-off point to consider low physical activity in this study.

Why physical activity level was not used as a cofounder? The author have this information.

Response: We are sorry for confusion. Low physical activity was one of the CHS frailty criteria. During the original clinical trial, participants were asked to maintain similar physical activity, and there were no significant changes in physical activity. Thus, the present study was not considered the low physical activity as a confounder.

Include the handgrip strength cut-off point to consider low handgrip strength. /was it the same cut-off used in the CHS. If yes, include the reference.

Response: Sorry for the mistake. We used the cut-off in the CHS and added the reference number “[4]”.

Why was cognitive impairment and frailty selected as cofounding factors and included in the fully adjusted model? P value for those variables was not < 0.20

Response: We are sorry for confusion. In the present study, confounding factors were assessed in the multivariate models and selected using p-value < 0.20 which was preferable for confounder selection by Greenland et al. (Annu Rev Public Health; 2015:89-108).

This secondary data analysis aimed to assess the amount of protein from habitual protein intake required to improve muscle mass and gait speed for older women and men. For this purpose, the participants were divided in six groups according sex and the changes in protein intake. How come the baseline characteristics of the participants were described and analyzed only by sex, irrespective of the changes in protein intake.

The characteristics and the differences of the participants in the six groups should be described here.

In the way it was described we don't even know how many participants there are in each of the six groups.

The analysis should consider the groups divided by tertiles of changes in protein intake and not divided only by sex.

Response: Thank you for your comments. We reanalyzed the baseline characteristics according to tertiles of change in protein intake in total older adults, women, and men separately in Table 1.

This description and the differences should also consider the six groups of the study.

Response: Thank you for your comments. We reanalyzed the changes in muscle mass and gait speed according to tertiles of change in protein intake in total older adults, women, and men separately in Table 2.

How were these cofounders defined?

Response: Sorry for the confusion. In the multivariate models, we selected confounding factors using p-value < 0.20 which was preferable for confounder selection by Greenland et al. (Annu Rev Public Health; 2015:89-108) in line 128-129.

Gait speed should be described in the baseline charactheristics of the participants.

Response: Sorry for the mistake. We added the baseline gait speed in Table 1.

in older men.

Response: Sorry for the mistake. We added the “in older men” in line 252.

Reviewer 3 Report

This article is the second analysis of a randomized controlled study to assess the amount of protein from habitual protein intake required to improve muscle mass and gait speed for older men and women. The topic is of interest and the manuscript is well written. However, there are a few critical problems to be clarified or revised as follows. 

  • The sample size is too small to draw any definite conclusions. Especially, the patient number of each group in men is around 10.
  • How did the authors calculate sample size in this randomized controlled study?
  • Please show the exercise protocol for all participants. Could all participants complete the exercise protocol during 12-week?
  • Please discuss why change in protein intake was not associated with change in muscle mass in older women more in detail.

Author Response

The sample size is too small to draw any definite conclusions. Especially, the patient number of each group in men is around 10.

Response: We agree with your opinions. Sample size of older men according to tertiles of changes in protein intake was small in absolute terms. However, power of ASM/weight, ASM/BMI, and ASM:fat ratio was ≥70% which was required to achieve reliability of statistical power by Cohen (Statistical power analysis for the behavioral sciences;1998).

How did the authors calculate sample size in this randomized controlled study?

Response: We are sorry for confusion. In our original clinical trial, sample size was calculated based on the findings of Candoew et al. (Med Sci Sports Exerc; 2008:1645-52), considering the mean ± SD increased in lean tissue mass of 3.2 ± 1.9 kg in the protein supplements group and 2.1 ± 1.4 kg in the nonprotein supplement group, with a power of 80%. Thus, this gave a sample size of 30 participants/group. With an expected dropout rate of 25%, a sample size of 40 participants/group was considered adequate.

Please show the exercise protocol for all participants. Could all participants complete the exercise protocol during 12-week?

Response: We are sorry for confusion. We stated the sentence “In addition, participants were asked to maintain their usual physical activity during 12-week intervention.” In line 68-69.

Please discuss why change in protein intake was not associated with change in muscle mass in older women more in detail.

Response: Thank you for your suggestions. We revised the discussion stating that women has lower baseline muscle mass and protein intake than men, and previous studies suggested women might be needed more protein to increase muscle mass than men in line 227-229.

Reviewer 4 Report

This is an interesting secondary analysis of a randomized controlled trial, showing that, in older frail males, an increased absolute daily protein intake above 0.54 g/kg/day is necessary to improve muscle mass. This effect was instead not observed in older frail females, suggesting gender-specific responses to protein supplementation in older age. 

I have the following comments: 

1) Criteria for definition of frailty and pre-frailty should be made explicit in the materials and methods section. 

2) How was exclusion for kidney failure defined? Was a GFR threshold used for including or excluding subjects? 

3) Did participants maintain their habitual exercise levels during the study? How were these levels assessed? 

4) The lack of association of variation in protein intake with gai speed is a critical point limiting the clinical relevance of the findings of this study. From a clinical perspective, it is much more important to improve functional parameters of older patients rather than improving muscle mass alone. 

5) The study population included a much larger number of females than males, and the sample size was small in absolute terms. This is a relevant limitation for the secondary analysis presented in this manuscript, whose results should not be overlooked.  

6) The circumstance that all participants were of Asian ethnicity may represent a limitation for the generalizability of results. 

7) The role of physical exercise as a countermeasure for sarcopenia and physical frailty, in combination with nutrition, should be more emphasized in the discussion. 

Author Response

1) Criteria for definition of frailty and pre-frailty should be made explicit in the materials and methods section. 

Response: Thank you for your suggestion. In line 85-86, we stated the definition of prefrail and frailty.

2) How was exclusion for kidney failure defined? Was a GFR threshold used for including or excluding subjects?

Response: We are sorry for confusion. We excluded older adults who were diagnosed with kidney failure by medical doctor.

3) Did participants maintain their habitual exercise levels during the study? How were these levels assessed?

Response: We are sorry for confusion. Participants were asked to maintain their usual physical activity during 12-week intervention, and there were no significant changes in physical activity during the study . 

4) The lack of association of variation in protein intake with gai speed is a critical point limiting the clinical relevance of the findings of this study. From a clinical perspective, it is much more important to improve functional parameters of older patients rather than improving muscle mass alone.

Response: Thank you for your suggestion. Previous studies reported that changes in muscle mass were not always change muscle function. We revised the discussion stating the results from previous studies in line 236-238.

5) The study population included a much larger number of females than males, and the sample size was small in absolute terms. This is a relevant limitation for the secondary analysis presented in this manuscript, whose results should not be overlooked.

Response: Thank you for your suggestion. We included small sample size in limitation in line 262.

6) The circumstance that all participants were of Asian ethnicity may represent a limitation for the generalizability of results.

Response: Thank you for your suggestion. We revised the limitation by stating that the generalizability of our findings to other populations may be limited since the participants of the present study included only undernourished frail Korean older adults line 266.

7) The role of physical exercise as a countermeasure for sarcopenia and physical frailty, in combination with nutrition, should be more emphasized in the discussion.

Response: Thank you for your suggestion. We revised the discussion that a combination of exercise and amino acid supplementation improved muscle mass and function compared with exercise or amino acid supplementation alone in sarcopenic older women [29].” in line 173-175.

Round 2

Reviewer 1 Report

The authors fully answered my questions.

Reviewer 2 Report

The quality of the paper has improved significantly.

Reviewer 3 Report

The authors have satisfactorily provided answers to all the questions I had raised during initial review. The paper is well revised and worthy for future publication for Nutrients.